# COVID-IRS: A novel predictive score for risk of invasive mechanical ventilation in patients with COVID-19

**José Antonio Garcia-Gordillo**[1], **Antonio Camiro-Zúñiga**[ID][1]\*, **Mercedes Aguilar-Soto**[ID][1], **Dalia Cuenca**[1], **Arturo Cadena-Fernández**[ID][1], **Latife Salame Khouri**[1], **Jesica Naanous Rayek**[1], **Moises Mercado**[ID][2], **The ARMII Study Group**[¶]

**1** Internal Medicine Department, The American British Cowdray Medical Center, Mexico City, Mexico, **2** Research Unit in Endocrine Diseases, Hospital de Especialidades, Centro Médico Nacional Siglo XXI, Instituto Mexicano del Seguro Social, Mexico City, Mexico

¶ ARMII: Asociación de Residentes de Medicina Interna en Investigacion (Research Association of Internal Medicine Residents), the complete membership of the author group is listed in the Acknowledgments.
\* antoniocamiro@hotmail.com

## Abstract

### Background

Coronavirus disease 2019 (COVID-19) is a systemic disease that can rapidly progress into acute respiratory failure and death. Timely identification of these patients is crucial for a proper administration of health-care resources.

### Objective

To develop a predictive score that estimates the risk of invasive mechanical ventilation (IMV) among patients with COVID-19.

### Study design

Retrospective cohort study of 401 COVID-19 patients diagnosed from March 12, to August 10, 2020. The score development cohort comprised 211 patients (52.62% of total sample) whereas the validation cohort included 190 patients (47.38% of total sample). We divided participants according to the need of invasive mechanical ventilation (IMV) and looked for potential predictive variables.

### Results

We developed two predictive scores, one based on Interleukin-6 (IL-6) and the other one on the Neutrophil/Lymphocyte ratio (NLR), using the following variables: respiratory rate, SpO2/FiO2 ratio and lactic dehydrogenase (LDH). The area under the curve (AUC) in the development cohort was 0.877 (0.823–0.931) using the NLR based score and 0.891 (0.843–0.939) using the IL-6 based score. When compared with other similar scores developed for the prediction of adverse outcomes in COVID-19, the COVID-IRS scores proved to be superior in the prediction of IMV.

**Data Availability Statement:** All files are available from the base_covid_20200918.xlsx databases in

https://www.kaggle.com/camiro/armii-study-group/version/1.

**Funding:** The author(s) received no specific funding for this work.

**Competing interests:** The authors have declared that no competing interests exist.

**Abbreviations:** SARS-CoV2, severe acute respiratory syndrome coronavirus 2; COVID-19, coronavirus disease 2019; ARDS, Acute Respiratory Distress Syndrome; IMV, Invasive Mechanical Ventilation; ICU, Intensive Care Unit; PCR, Polymerase Chain Reaction; CT, Computed Tomography; CBC, Complete blood count; ROC, Receiver Operator Curve; CRP, C Reactive Protein; LDH, Lactate Dehydrogenase; IL-6, Interleukin 6; NLR, Neutrophil/Lymphocyte Ratio.

## Conclusion

The COVID-IRS scores accurately predict the need for mechanical ventilation in COVID-19 patients using readily available variables taken upon admission. More studies testing the applicability of COVID-IRS in other centers and populations, as well as its performance as a triage tool for COVID-19 patients are needed.

## Background

SARS-CoV2 is a viral pathogen that causes coronavirus disease 2019 (COVID-19). The clinical spectrum of COVID-19 varies widely. Up to 80% of patients present with an inconsequential flu-like illness, but 20% develop a form of viral pneumonia with acute respiratory distress syndrome (ARDS). In turn, 15% require support with invasive mechanical ventilation (IMV) [4–6]. Among hospitalized COVID-19 patients, 5–33% will require admission to an intensive care unit (ICU) and 75% to 100% of them will require IMV [1]. Mortality rates vary from center to center, but in general they remain high in the group of critically ill patients who develop respiratory failure and require admission to ICU for IMV [5].

Since the original outbreak in Wuhan, China in December 2019, SARS-CoV2 has rapidly spread around the world reaching unprecedented pandemic proportions and overwhelming healthcare systems worldwide [2]. Mexico's public health system represents one of those cases, being the country with the third highest COVID-19 mortality rate [3, 4]. One of the main challenges of the COVID-19 pandemic has been performing a proper triage that allows reasonable and cost-effective allocation of health-care resources [5–7]. Identifying patients that are likely to evolve into severe disease is a challenging task that surpasses good clinical judgement. Thus, there is an urgent need to develop tools capable of predicting the course of the disease. These could aid clinicians to select patients who are at risk and therefore warrant early life-saving interventions [4].

### Objectives

To develop a new severity score for the prediction of IMV in COVID-19.

### Study design

We retrospectively collected information from all COVID-19 patients aged 18 years or older admitted to the American British Cowdray Medical Center, a private teaching hospital in Mexico City, between March 12 and August 10, 2020. The diagnosis of COVID-19 was suspected based on clinical manifestations and confirmed by means of a positive PCR for SARS-CoV-2, which was carried out according to the Centers for Disease Control published guidelines [8] or in case of a negative PCR, with a chest CT scan with characteristic findings for COVID-19. The primary outcome was the need for IMV.

Exclusion criteria included having a "Do Not Resuscitate" order or having incomplete data in the electronic medical record. The ethics committee waived the requirement for an informed consent. All the analyzed data was fully anonymized from the moment it was captured and remained so during the entire duration of the study. The protocol (ID: ABC-20-50) was approved by our local scientific and ethics committees (*Comité de Ética en Investigación*, American British Cowdray Medical Center) and conducted according to the principles of the Helsinki declaration.

### Development and validation cohort election

We divided the cohort in two groups of roughly equal size using a random number generation algorithm. The larger group was used for the development cohort, while the smaller group was used as the validation cohort. We compared both cohorts using the chi-square test for categorical variables and Man-Whitney U test for continuous variables, in order to find significant differences in their baseline characteristics and outcomes.

### Potential predictive variables

We categorized patients' characteristics at hospital admission into the following groups of variables: demographic and anthropometric characteristics, clinical features, medical history, laboratory results, and clinical outcomes. Demographic and anthropometric characteristics included age, gender, body mass index, and ethnicity. Clinical features included vital signs, presence of symptoms characteristic of COVID-19 (dyspnea, fever, cough, etc.), and date of symptom onset. Medical history included currently diagnosed comorbidities (diabetes, hypertension, cancer, etc.), smoking status, alcohol consumption, and current medical treatments. Laboratory results included complete blood count (CBC), coagulation tests, blood chemistry panel, liver function tests, lipid profile, inflammatory markers, including interleukin-6 (IL-6), ultrasensitive C reactive protein (CRP), D-dimer, fibrinogen and procalcitonin, as well as and 25-hydroxi-vitamin D3. Clinical outcomes included in-hospital death, length of stay and the need for invasive mechanical ventilation (IMV).

### Predictive variable selection

Using the development cohort, we performed univariate logistic regressions for IMV using all the variables mentioned above. We selected all variables that had a p value <0.1 and conducted a backwards stepwise multivariate logistic regression to find the variables that were independently associated with the requirement of IMV. After the selection of the optimal variables for the model, in order to ensure the model's applicability in most settings, we checked for the laboratory variable's availability in general settings. This was done via a telephonic interview on 7 different general hospitals in Mexico City and its surroundings. The variables that were not available in more than 50% of the screened hospitals were deemed to be not readily available. We tested for similar variables using the Spearman correlation test in order to identify suitable surrogates. Thus, we developed two predictive models, one constructed with optimal variables and the other one with accessible surrogate variables.

### Construction of the score and assessment of accuracy

After identifying the predictive variables, we carried out locally weighted scatterplot smoothing (LOWESS) curves on numerical variables in order to determine adequate intervals and cut-off points on both models. Subsequently, in order to assign a scoring value to the selected variables, we estimated their coefficient of variation using univariate logistic regressions and assigned the rounded-up coefficient as the numeric value for the score in the corresponding strata. We constructed receiver operating characteristic (ROC) curves in order to evaluate the performance of our scores. Evaluation for goodness of fit was carried out by means of the Hosmer-Lemeshow test and predictive performance was ascertained by the concordance index (C-index). We evaluated internal calibration with 2000 bootstrap samples. The score underwent external validation by comparing the ROC curves of the development and validation cohorts. Finally, we compared the ROC curves of our score with the calculated ROC curves of other scores that predict ventilatory deterioration or other adverse outcomes in COVID-19 patients (ABC-GOALScl,

COVID-GRAM, NEWS-2, CURB-65, and CALL prediction model) [9–13] in both, the development and validation cohorts. We compared the ROC curves of the aforementioned scores using only the data from those patients in whom all the scores were calculated appropriately. We performed all statistical analyses using STATA version 14 (StataCorp, College Station, Texas, USA) and GraphPad Prism 6.0 (GraphPad Software, San Diego, CA, USA).

## Results

The score development cohort comprised 211 patients (52.62% of total sample) whereas the validation cohort included 190 patients (47.38% of total sample). We divided participants according to the need of IMV. Baseline population characteristics are depicted in Table 1. The comparison between the development and validation cohorts is shown in S1 Table (S1 Table. Comparison between the development and validation cohorts).

### Predictive variables selection and score construction

S2 Table (S2 Table. Univariate logistic regressions for variable selection) depicts the univariate logistic regressions for all individual variables. Based on the backwards stepwise multivariate logistic regression (S3 Table. Multivariate logistic regression), we selected the following predictive variables for the development of the score: Respiratory rate, SpO2/FiO2 ratio, LDH and IL-6. Since IL-6 was deemed as not readily available in most settings, we decided to use the Neutrophil/Lymphocyte Ratio (NLR) as a suitable surrogate, due to its easy availability and good performance in both, the correlation test (Spearman's rho = 0.485, p<0.001) and the multivariate logistic model (coefficient 0.049, p = 0.004, R-squared = 0.3428) (Fig 1) (S4 Table. Spearman's correlation results and R-squared of multivariate logistic regression models for surrogate variables).

We named our score COVID-IRS (Intubation Risk Score). We constructed two different versions of the score: COVID-IRS-IL6 using the optimal model and COVID-IRS-NLR using the accessible variables. We further stratified the aforementioned scores into low, moderate, high, and very high-risk categories. The scores and their respective interpretations are shown in Fig 2. Although there was a tendency towards a higher median amount of days between patient admission to the hospital and the requirement of IMV in lower risk groups (ex. 5 days in low risk patients vs. one day in high risk patients) these differences did not prove to be statistically significant (COVID-IRS-NLR, p = 0.371; COVID-IRS-IL6, p = 0.275) (S1 Fig. Median days from patient admission until IMV requirement by risk group).

### Assessment of accuracy

Fig 3 shows the ROC curves for both scores in the development and validation cohorts. The area under the curve (AUC) in the development cohort was 0.877 (0.823–0.931) using the NLR based score and 0.891 (0.843–0.939) using the IL-6 based score. Internal validation was excellent, with the goodness-of-fit tests being statistically significant (NLR: p = 0.179; IL-6 p = 0.189), as well as the bootstrap replications (NLR: p<0.001; IL-6 p<0.001). The AUC in the validation cohort was smaller than the one in the development cohort, with 0.823 (0.758–0.887) using the NLR based score and 0.826 (0.759–0.892) using the IL-6 based score. A good correlation was found between predicted and measured risks (S2 Fig. Predicted and observed percentages of patients who required IMV at each point of both COVID-IRS scores in the development and validation cohorts.). Optimal cutoff points in the validation cohort for the COVID-IRS-NLR score and the COVID-IRS-IL6 were >6 (S: 68.57%, E: 87.5%) and >5 (S: 72.86%, E: 81.67%). Table 2 depicts the comparison between the AUC of all scores. When

**Table 1. Baseline characteristics of included patients.**

|  | IMV (n = 142) | No IMV (n = 259) | p-value |
|---|---|---|---|
| Age, years | 57.95 (49.22–67.29) | 50.5 (40.74–65.1) | <0.001 |
| Male sex | 107 (75.35) | 157 (60.6) | 0.003 |
| BMI | 28.09 (25.95–32.52) | 27.54 (25.09–31.16) | 0.075 |
| Tabaquic index | 6 (2–20) | 2.45 (0.5–15) | 0.106 |
| Diabetes | 30 (21.13) | 38 (14.72) | 0.075 |
| Hypertension | 52 (36.61) | 66 (25.48) | 0.011 |
| COPD | 5 (3.52) | 3 (1.15) | 0.096 |
| CKD | 3 (2.11) | 5 (1.93) | 0.874 |
| Vital signs on admission | | | |
| Cardiac rate | 83 (75–90) | 80 (73–88) | 0.064 |
| Respiratory rate | 24 (20–30) | 20 (18–22) | <0.001 |
| Mean arterial pressure | 86.16 (79–90) | 86.66 (81.66–93) | 0.100 |
| Oxygen saturation | 85 (76–90) | 91 (88–94) | <0.001 |
| SaO2/FiO2 ratio | 94 (80–155) | 225 (210–237) | <0.001 |
| Temperature | 36.4 (36–37) | 36.4 (36–37) | 0.750 |
| Days until admission* | 8 (5–13) | 8 (6–11) | 0.55 |
| Laboratory values | | | |
| Hemoglobin | 14.8 (13.5–15.9) | 14.8 (13.5–16.2) | 0.689 |
| Leucocytes | 8.9 (6.4–12.4) | 6.7 (5.2–8.9) | <0.001 |
| Lymphocytes | 905 (610–1170) | 1040 (780–1480) | <0.001 |
| Neutrophils | 7070 (4960–10510) | 4810 (3300–6780) | <0.001 |
| NLR | 8 (5.27–13.27) | 4.46 (2.73–6.96) | <0.001 |
| Platelets | 218 (161–274) | 215 (173–285) | 0.785 |
| HbA1c | 6.2 (5.9–7.5) | 5.8 (5.4–6.3) | <0.001 |
| D-dimer | 1089 (649–1764) | 753 (485–1154) | <0.001 |
| INR | 1.035 (0.94–1.09) | 0.97 (0.92–1.04) | 0.014 |
| Fibrinogen | 400 (323–547) | 448 (364–561) | 0.074 |
| Albumin | 3.42 (3–3.74) | 3.89 (3.63–4.2) | <0.001 |
| AST | 42 (28.8–66.4) | 33.25 (21.1–49) | <0.001 |
| ALT | 37 (24–65) | 33.5 (21–53) | 0.072 |
| ALP | 86.5 (65–112) | 78 (64–101) | 0.116 |
| GPT | 81 (58–152) | 83 (41–115) | <0.001 |
| TB | 0.59 (0.4–0.8) | 0.46 (0.33–0.67) | 0.017 |
| Glucose | 126 (108–163) | 108 (97–126) | <0.001 |
| BUN | 18.1 (13.9–25.1) | 13.2 (10.3–17.6) | <0.001 |
| Creatinine | 0.95 (0.78–1.17) | 0.85 (0.72–1.02) | 0.002 |
| CPK | 118 (58–290) | 89 (53–182) | 0.0146 |
| LDH | 371 (287–441) | 257 (203–324) | <0.001 |
| C Reactive Protein | 18.37 (9.44–29.57) | 7.35 (2.92–14.48) | <0.001 |
| Procalcitonin | 0.36 (0.13–1.03) | 0.12 (0.06–0.22) | <0.001 |
| Ferritin | 1334 (849–2378) | 683 (289–1301) | <0.001 |
| IL-6 | 143 (54–232) | 47.3 (19.4–91.8) | <0.001 |
| IgG | 1070 (912–1270) | 1127 (980–1315) | 0.199 |
| IgM | 82.9 (62.1–134.2) | 97 (69–136) | 0.145 |
| Death | 27 (19.01) | 3 (1.16) | <0.001 |
| Length of stay | 19 (14–26) | 7 (5–9) | <0.001 |
| COVID-19 treatment given | | | |

*(Continued)*

**Table 1.** (Continued)

|  | **IMV (n = 142)** | **No IMV (n = 259)** | **p-value** |
|---|---|---|---|
| Lopinavir/ritonavir | 99 (69.91) | 146 (56.36) | 0.016 |
| Azithromycin | 122 (85.96) | 180 (69.72) | 0.001 |
| Hydroxychloroquine | 118 (83.19) | 195 (76.15) | 0.139 |
| Tocilizumab | 105 (74.04) | 71 (27.54) | <0.001 |
| Corticosteroids | 121 (85.11) | 169 (65.23) | <0.001 |

Values are percentages or median (IQR) as appropriate. IMV: Invasive Mechanical Ventilation, BMI: Body Mass Index, COPD: Chronic Obstructive Pulmonary Disease, CKD: Chronic Kidney Disease, SaO2: Oxygen saturation, FiO2: Fraction of inspired oxygen, NLR: Neutrophil/Lymphocyte Ratio, INR: International Normalized Ratio, AST: Aspartate Aminotransferase, ALT: Alanine Aminotransferase, ALP: Alkaline Phosphatase, GPT: Glutamic Pyruvic Transaminase, TB: Total Bilirubin, BUN: Blood Urea Nitrogen, CPK: Creatinine Phosphokinase, LDH: Lactate Dehydrogenase, IL-6: Interleukin 6, IgG: Immunoglobulin G, IgM Immunoglobulin M. *Days from symptom onset until hospital admission

compared to other scores, the AUC of both COVID-IRS scores was superior to that shown by all other calculated risk scores in both the development and validation cohorts.

## Discussion

In this study, we developed two novel prognostic scores for the prediction of IMV requirement in COVID-19 patients, using variables registered upon hospital admission. ROC analysis of data derived from both the development and the validation cohorts revealed an excellent

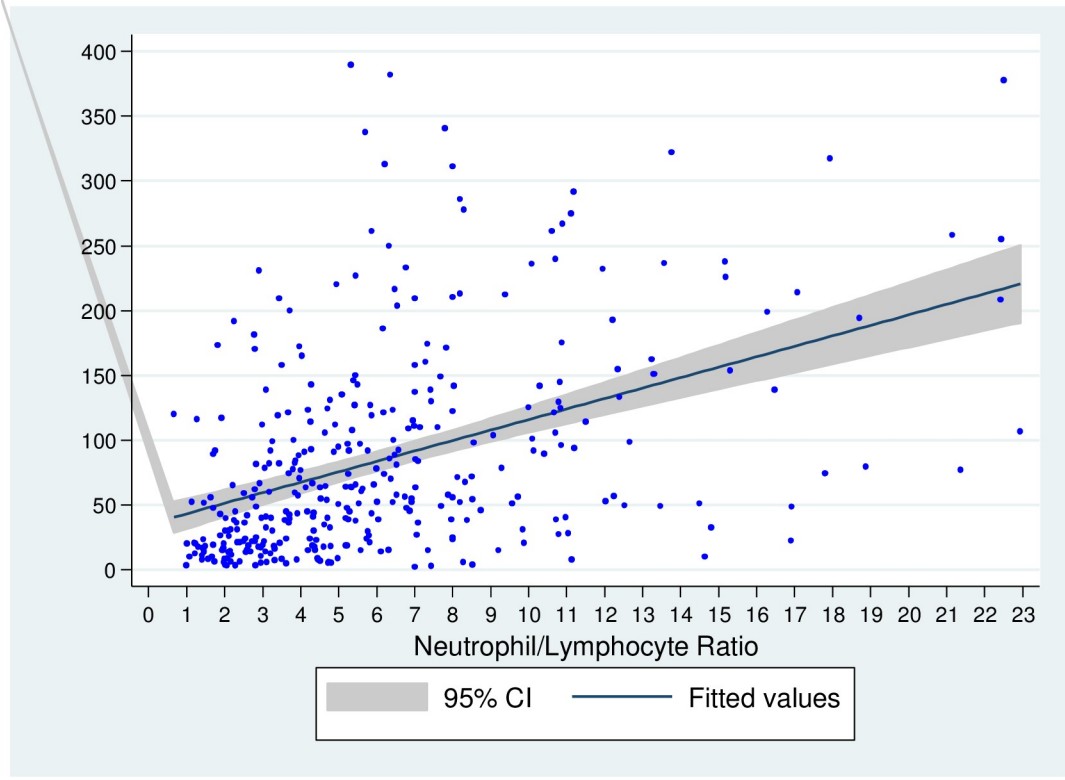

**Fig 1. Correlation between IL-6 and NLR.** Here we show the correlation between NLR and IL-6. The correlation produced a Spearman's rho of 0.485, which was statistically significant with a p value of <0.001. Dots represent individual values.

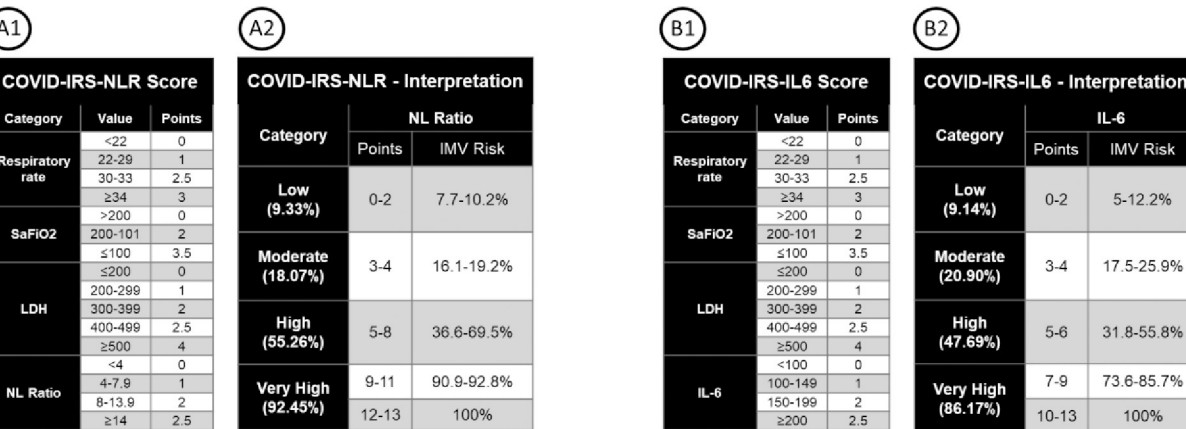

**Fig 2. COVID-IRS-NLR and COVID-IRS-IL6 scoring and interpretation.** Here we show the algorithm for calculating both COVID-IRS-NLR (A1 and A2) and COVID-IRS-IL6 (B1 and B2) scores. Scores are assigned using the cut points in either the A1 or B1 panel, and the resulting sum is interpreted in the corresponding A2 or B2 panel, which in turn dictates the risk strata for IMV.

performance of the NLR-based as well as of the IL-6-based scores. Importantly, according to our analysis, the NLR proved to be an outstanding surrogate of IL-6. When compared with other similar scores developed for the prediction of adverse outcomes in COVID-19, the COVID-IRS scores proved to be superior in the prediction of IMV.

We believe that the biomarkers used in the COVID-IRS scores (respiratory rate, SaO2/FiO2 ratio, LDH, and either IL-6 or NLR), accurately represent relevant aspects of the clinical phenomena seen in severe COVID-19. Both, the respiratory rate and the SaO2/FiO2 ratio evaluate ventilatory function, whose deterioration is the main component associated with COVID-19 mortality [9, 10]. The SaO2/FiO2 ratio was used as a surrogate for the PaO2/FiO2 ratio due to its availability and because it maintains a close linear relationship with O2-CO2 exchange and blood oxygenation [11]. LDH is involved in the anaerobic metabolism of glucose and thus, is upregulated when oxygen supplies are limited [12]. LDH levels are increased in patients with COVID-19 pneumonia and have been associated with adverse outcomes and consistently included in COVID-19 severity scores [12]. Finally, IL-6 and the NLR reflect the severity of the ongoing inflammatory process and immune dysregulation [13–16]. IL-6 is a pleiotropic cytokine mainly secreted by activated macrophages in response to any aggressor. It promotes the production of acute phase reactants and the proliferation of myeloid cells, as well as neutrophil survival in lung tissue [17, 18]. On the other hand, neutrophils as effectors of the innate immune system may reflect the severity of pneumonia and have been used as markers of poor prognosis in different inflammatory states, such as sepsis [17]. Lymphocytes, another important cell of the immune system, are recruited to damaged tissues and in the context of COVID-19 tend to migrate to lung and blood vessels, which partially accounts for the low peripheral lymphocyte count seen in these patients [19, 20]. Thus, a high NLR is a reflection of the severity of the ongoing inflammatory process [21–23].

Both IL-6 and NLR have been used as prognostic markers in both, influenza and community-acquired pneumonia [24]. It therefore seemed logical to try to use them as predictive biomarkers in patients with SARS-Cov-2 pneumonia [24, 25]. Since the beginning of the pandemic leukocytosis, lymphopenia and high levels of IL-6 have been consistently associated with poor prognosis in patients with COVID-19 infection [25]. The correlation between NLR and IL-6 has been previously described in other clinical contexts [11, 18]. Our study is perhaps the first one to evaluate the equivalency between the NLR and the serum levels of IL-6 in the context of

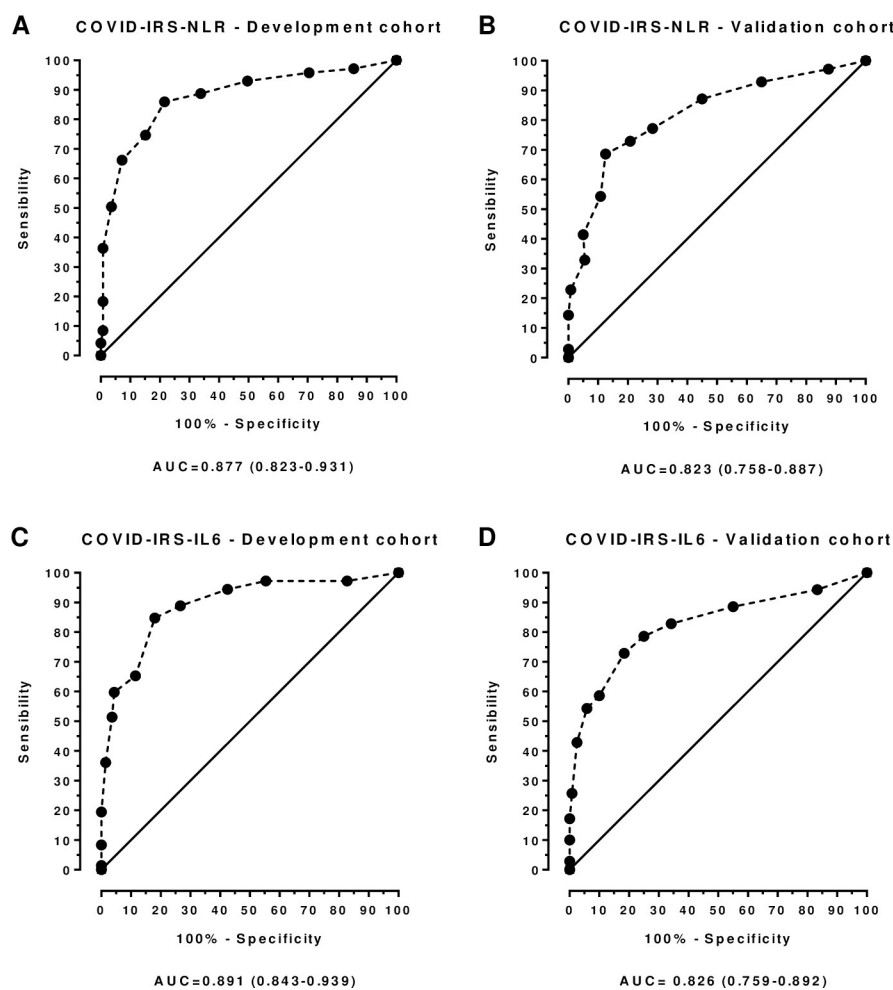

**Fig 3. AUC comparison between development and validation cohorts in both scores.** Here we show the comparison of both scores AUC between the development and validation cohorts. Panels A and C show COVID-IRS-NLR's AUC for the development and validation cohorts, which were measured at 0.877 (0.823–0.931) and 0.823 (0.758–0.887). In turn, panels B and D show COVID-IRS-IL6's AUC for the development and validation cohorts, which were measured at 0.891 (0.843–0.939) and 0.826 (0.759–0.892).

**Table 2. Comparison of AUC across different risk scores.**

| Score | Development cohort (N = 190) | Validation cohort (N = 172) | P value of comparison with COVID-IRS-NLR score | |
|---|---|---|---|---|
| | | | Development cohort | Validation cohort |
| COVID-IRS-NLR | 0.870 (0.809–0.931) | 0.850 (0.791–0.910) | - | - |
| COVID-IRS-IL6 | 0.883 (0.829–0.937) | 0.852 (0.788–0.916) | 0.249 | 0.783 |
| COVID-GRAM | 0.787 (0.719–0.855) | 0.773 (0.702–0.844) | 0.005 | 0.029 |
| ABC-GOALScl | 0.765 (0.698–0.831) | 0.739 (0.667–0.812) | 0.001 | 0.001 |
| PREDICO | 0.704 (0.630–0.778) | 0.791 (0.724–0.857) | <0.001 | 0.201 |
| NEWS2 | 0.723 (0.645–0.800) | 0.789 (0.714–0.864) | <0.001 | 0.037 |
| CALL | 0.679 (0.606–0.753) | 0.678 (0.602–0.755) | <0.001 | <0.001 |
| CURB-65 | 0.739 (0.665–0.812) | 0.709 (0.629–0.789) | <0.001 | 0.002 |
| SOFA | 0.888 (0.839–0.937) | 0.862 (0.806–0.917) | 0.770 | 0.291 |

All values are expressed as AUC (95% Confidence Interval)

COVID-19 severity. Even though both measurements seem to accurately reflect severity, IL-6 measurements require specialized equipment and are only readily available in few centers, while the NLR only requires a CBC, which is inexpensive and widely available [19, 20].

Different prognostic scores for COVID-19 have been developed using different variables, including the presence of comorbidities, age, absolute lymphocyte count, LDH, oxygen saturation, respiratory rate, and bilateral opacities on CT scan in order to identify patients at risk of adverse outcomes [26–30]. There are some predictive scores with similar applications to the COVID-IRS score. The COVID-GRAM score was created to calculate the probability of developing critical COVID-19 using data from 1590 Chinese patients. The AUC on both the development and the validation cohorts were 0.88 [27]. Another score is the ABC-GOALS, developed to predict ICU admission, and is based on data from 329 patients admitted to a COVID-19 reference center in Mexico City. The ABC-GOALS score has 3 versions, a clinical only model (ABC-GOALSc), a clinical and laboratory model (ABC-GOALScl), and a clinical, laboratory and x-ray model (ABC-GOALSclx). We only compared our data with the ABC-GOALScl score, due to our lack of more precise CT scan interpretation data in our dataset. The AUC of the ABC-GOALScl score was 0.86 and 0.87 in its development and validation cohorts, respectively. More recently the PREDICO score has been developed for the prediction of severe respiratory failure, using the data of 1265 patients from eleven Italian hospitals. The AUC was of 0.89 and 0.85 in its development and validation cohorts. All the aforementioned scores have several variables in common with the COVID-IRS score like LDH, Lymphocyte count (NLR in the COVID-GRAM score), respiratory rate and SaO2/FiO2 ratio [29]. Even though both these scores were not developed for the specific identification of patients that were going to require IMV, they achieved lower AUC when they were tested directly in our population, in both the development and validation cohorts (COVID-GRAM: 0.787 and 0.773; ABC-GOALScl: 0.765 and 0.739). As mentioned earlier, both COVID-IRS was superior to the COVID-GRAM and ABC-GOALScl scores at predicting the need for IMV. Additionally, the Brescia-COVID Respiratory Severity Scale (BCRSS), a stepwise approach to managing patients with confirmed/presumed COVID-19 pneumonia [31], is a meaningful tool based on clinical features and chest x-ray changes, for determining the scalation in ventilatory support. It is meant to be dynamic and frequently reassessed and re-scored after interventions and has been widely used in that center for evaluating patients from de emergency department and throughout hospitalization. We weren't able to estimate and compare the BCRSS's performance in our cohort to predict the IMV risk, due to lack of information in our records. Finally, all variables needed to calculate the COVID-GRAM, ABC-GOALScl, PREDI-CO and COVID-IRS-NLR scores can be easily obtained in the outpatient setting and could complement each other. Of important note the SOFA score had a similar AUC when compared with the COVID-IRS scores for predicting IMV. Due to the retrospective nature of our data, we did not distinguish patients who needed IMV on arrival or first day of admission from those who were intubated during their hospital stay, and when taking into consideration that the SOFA score includes a variable for IMV, this most likely results in an overestimation of its capacity to predict the need for IMV in our population.

It is important to emphasize that some high-risk patients may not present with signs of respiratory distress upon admission, but can rapidly progress to ARDS, and thus need frequent monitoring [9, 29, 30]. In order to avoid overwhelming of health care systems worldwide, the identification of these patients is a priority. The timely identification of these cases could help to reduce mortality and allow a reasonable and cost-effective allocation of human resources and infrastructure [5, 31]. One of the possible benefits of our score, comes from its utility in identifying which patients require this closer surveillance and which can have their evaluations spaced-out safely. We identified four risk categories according to the probability of requiring IMV: low, moderate, high and very high risk. Low-risk patients have a low probability of

requiring IMV and could benefit from a strategy that offers early discharge from the hospital and subsequent ambulatory visits. Patients with moderate-risk scores could remain in a hospital ward for surveillance. Finally, the high-risk and very high-risk category patients have an IMV probability of over 31.8%, and could therefore should be kept in a ward that has enough personnel to provide frequent re-evaluations and prompt response times for emergency endotraqueal intubation (like intermediate care units). Further studies are needed in order to validate this application of the COVID-IRS.

The main limitations of our study are its retrospective nature and the fact that some of the patients received different medical treatments prior to hospitalization (such as glucocorticoids) which could act as confounders. Our results may not be representative of the general real-life situation prevailing in most COVID-19 centers; our mortality rate is rather low, which can be attributed to the availability of ICU facilities. Finally, the incidence of comorbidities and old age in our cohort is lower than that reported in other centers and could thus prove to be a factor that hampers its application in other settings.

## Supporting information

**S1 Table. Comparison between the development and validation cohorts.** Values are percentages or median (IQR) as appropriate. IMV: Invasive Mechanical Ventilation, BMI: Body Mass Index, COPD: Chronic Obstructive Pulmonary Disease, CKD: Chronic Kidney Disease, SaO2: Oxygen saturation, FiO2: Fraction of inspired oxygen, NLR: Neutrophil/Lymphocyte Ratio, INR: International Normalized Ratio, AST: Aspartate Aminotransferase, ALT: Alanine Aminotransferase, ALP: Alkaline Phosphatase, GPT: Glutamic Pyruvic Transaminase, TB: Total Bilirubin, BUN: Blood Urea Nitrogen, CPK: Creatinine Phosphokinase, LDH: Lactate Dehydrogenase, IL-6: Interleukin 6, IgG: Immunoglobulin G, IgM Immunoglobulin M. (DOCX)

**S2 Table. Univariate logistic regressions for variable selection.** BMI: Body Mass Index, COPD: Chronic Obstructive Pulmonary Disease, CKD: Chronic Kidney Disease, SaO2: Oxygen saturation, FiO2: Fraction of inspired oxygen, NLR: Neutrophil/Lymphocyte Ratio, INR: International Normalized Ratio, AST: Aspartate Aminotransferase, ALT: Alanine Aminotransferase, ALP: Alkaline Phosphatase, GPT: Glutamic Pyruvic Transaminase, TB: Total Bilirubin, BUN: Blood Urea Nitrogen, CPK: Creatinine Phosphokinase, LDH: Lactate Dehydrogenase, IL-6: Interleukin 6, IgG: Immunoglobulin G, IgM Immunoglobulin M. (DOCX)

**S3 Table. Multivariate logistic regression.** SaO2: Oxygen saturation, FiO2: Fraction of inspired oxygen, LDH: Lactate Dehydrogenase, IL-6: Interleukin 6, NLR: Neutrophil/Lymphocyte Ratio. (DOCX)

**S4 Table. Spearman's correlation results and R-squared of multivariate logistic regression models for surrogate variables.** NLR: Neutrophil/Lymphocyte Ratio. (DOCX)

**S1 Fig. Median days from patient admission until IMV requirement by risk group.** Here we show the median time in days from patient admission until the patients required the initiation of IMV. There was a tendency towards a higher median amount of days between patient admission to the hospital and the requirement of IMV in lower risk groups. These differences did not prove to be statistically significant (COVID-IRS-NLR, p = 0.371; COVID-IRS-IL6, p = 0.275). (TIF)

**S2 Fig. Predicted and observed percentages of patients who required IMV at each point of both COVID-IRS scores in the development and validation cohorts.** Here we show the correlation between observed and predicted percentages of patients who required IMV. Both predicted and measured risks showed a strong correlation.
(TIF)

## Acknowledgments

To all of our residents and friends from the ICU, for their amazing labor and commitment during the pandemic, which has allowed us to have a minute mortality rate. To all of the ARMII study group, who made this work possible: Isabel Gutiérrez-Lozano, Jorge Carlos Salado-Burbano, Rodolfo Jiménez-Soto, Mariana Vélez-Pintado, Alejandra Kerbel Laiter, Guillermo Bracamontes-Castelo, Cecilia Nehmad Misri, Carlos Andrés Rodríguez-Toledo, Alma Nelly Rodríguez-Alcocer, Mariana Rotzinger-Rodríguez, Stefany Jacob Kuttothara, Renzo Pérez-Dórame, Ana Paula Landeta-Sa, Mariana Covadonga Ansoleaga-García, Andrea Romo López, Santiago Montiel-Romero, José Carlos Krause Marún, Juan Pablo Guillermo-Durán, María Fernanda Coss-Rovirosa, Victor José Leal Alcántara, María Luisa Montes de Oca-Loyola, Adolfo Díaz Cabral, Laura Crespo-Ortega, Walter Valle-Uitzil, Rodrigo Sánchez Magallán, Issac O. Vargas Olmos, Víctor Hugo Gomez-Johnson, Gina Gonzalez Calderón, Tábata Cano-Gámez. Lead autor: Mercedes Aguilar-Soto–e-mail: mercedesaguilarsoto@gmail.com.

We would also like to deeply thank the support of Eduardo Fernandez Campuzano and the Internal Medicine group practice of the American British Cowdray Medical Center, whose guidance and support made this possible. This research did not receive any specific grant from funding agencies in the public, commercial, or not-for-profit sectors. Declaration of interests: none.

## Author Contributions

**Conceptualization:** José Antonio Garcia-Gordillo, Antonio Camiro-Zúñiga, Mercedes Aguilar-Soto, Dalia Cuenca, Moises Mercado.

**Data curation:** Antonio Camiro-Zúñiga, Mercedes Aguilar-Soto, Latife Salame Khouri.

**Formal analysis:** Antonio Camiro-Zúñiga, Mercedes Aguilar-Soto.

**Investigation:** José Antonio Garcia-Gordillo, Antonio Camiro-Zúñiga, Mercedes Aguilar-Soto, Dalia Cuenca, Arturo Cadena-Fernández, Latife Salame Khouri, Jesica Naanous Rayek.

**Methodology:** Antonio Camiro-Zúñiga, Mercedes Aguilar-Soto, Latife Salame Khouri.

**Project administration:** Antonio Camiro-Zúñiga, Mercedes Aguilar-Soto.

**Resources:** Antonio Camiro-Zúñiga.

**Software:** Antonio Camiro-Zúñiga.

**Supervision:** Antonio Camiro-Zúñiga, Mercedes Aguilar-Soto, Dalia Cuenca, Moises Mercado.

**Validation:** Antonio Camiro-Zúñiga, Mercedes Aguilar-Soto.

**Visualization:** Antonio Camiro-Zúñiga.

**Writing – original draft:** José Antonio Garcia-Gordillo, Antonio Camiro-Zúñiga, Mercedes Aguilar-Soto, Dalia Cuenca, Arturo Cadena-Fernández, Moises Mercado.

**Writing – review & editing:** José Antonio Garcia-Gordillo, Antonio Camiro-Zúñiga, Mercedes Aguilar-Soto, Dalia Cuenca, Latife Salame Khouri, Jesica Naanous Rayek, Moises Mercado.

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
