## [Decision Letter · Decision Letter 0]

22 Dec 2020

PONE-D-20-32632

COVID-IRS: a novel predictive score for risk of invasive mechanical ventilation in patients with COVID-19

PLOS ONE

Dear Dr. Camiro-Zuñiga,

Thank you for submitting your manuscript to PLOS ONE. After careful consideration, we feel that it has merit but does not fully meet PLOS ONE’s publication criteria as it currently stands. Therefore, we invite you to submit a revised version of the manuscript that addresses the points raised during the review process.

We look forward to receiving your revised manuscript.

Kind regards,

Antonio Palazón-Bru, PhD

Academic Editor

PLOS ONE

Journal Requirements:

2.Thank you for including your ethics statement: 'The protocol was approved by our scientific and ethics committees and conducted according to the principles of the Helsinki declaration.'   

(a) Please amend your current ethics statement to include the full name of the ethics committee/institutional review board(s) that approved your specific study.  

(b) Once you have amended this/these statement(s) in the Methods section of the manuscript, please add the same text to the “Ethics Statement” field of the submission form (via “Edit Submission”).

3. In ethics statement in the manuscript and in the online submission form, please provide additional information about the patient records/samples used in your retrospective study. Specifically, please ensure that you have discussed whether all data/samples were fully anonymized before you accessed them and/or whether the IRB or ethics committee waived the requirement for informed consent. If patients provided informed written consent to have data/samples from their medical records used in research, please include this information.

4. Please note that PLOS does not permit references to “data not shown.” Authors should provide the relevant data within the manuscript, the Supporting Information files, or in a public repository. If the data are not a core part of the research study being presented, we ask that authors remove any references to these data.

5. Please amend your list of authors on the manuscript to ensure that each author is linked to an affiliation. Authors’ affiliations should reflect the institution where the work was done (if authors moved subsequently, you can also list the new affiliation stating “current affiliation:….” as necessary).

6. One of the noted authors is a group or consortium [ARMII Study Group]. In addition to naming the author group, please list the individual authors and affiliations within this group in the acknowledgments section of your manuscript. Please also indicate clearly a lead author for this group along with a contact email address.

7. Please amend either the abstract on the online submission form (via Edit Submission) or the abstract in the manuscript so that they are identical.

Reviewers' comments:

Reviewer's Responses to Questions

**Comments to the Author**

1. Is the manuscript technically sound, and do the data support the conclusions?

Reviewer #1: Yes

Reviewer #2: Yes

2. Has the statistical analysis been performed appropriately and rigorously? 

Reviewer #1: N/A

Reviewer #2: Yes

3. Have the authors made all data underlying the findings in their manuscript fully available?

Reviewer #1: No

Reviewer #2: Yes

4. Is the manuscript presented in an intelligible fashion and written in standard English?

Reviewer #1: Yes

Reviewer #2: Yes

5. Review Comments to the Author

Reviewer #1: Garcia-Gordillo conducted a retrospective study to investigate the new models to predict the need of IMV in patients with COVID-19. Although both NLR and IL-6 were well recognized as indicators of worse outcome in patients with covid-19, this study provided some useful information for clinical practices.

I have some comments:

1) It’s problematic to use variables in the model at your own discretion which you describe in the method section. These variables should be determined base on standard statistic methods.

2) The authors described in the abstract that “When compared with other similar 42 scores developed for the prediction of adverse outcomes in COVID-19, the COVID-IRS 43 scores proved to be superior in the prediction of IMV.” However, no comparisons were done and no p values were given in Table 2. Moreover, more scores such as APACHE-Ⅱand SOFA should be compared with the models.

3) the optimal cutoff value of these models should be shown in order to provide more information for clinical practices.

4) This study needs to be reviewed by a statistician.

Reviewer #2: I am pleased to read a good work such this even if retrospective. I think some clarifications are needed in order to be published and in my opinion are minor revisions.

1) All the patients were evaluated to be, in case of need, "resuscitated"? I mean, It's important to clarify there was not a DNR order for some patients

2) If possible, in order to identify the inflammatory phase, I would put the variable "time since symptoms onset to ED admission".

3) All the patients came from community? nosocomial or health care related infections?

4) I would explain which treatment (corticosteroids, remdesivir, immunomodulators) have been given to the patients belonging to the 2 groups

5) If not intubated, I would explain which kind of ventilatory support was given (nasal flow cannula, venturimask, CPAP,BiPAP)

6) I woud cite in the discussion the Brescia-COVID Respiratory Severity Scale as the first score that was made worldwide out of China (Brescia has been, with Bergamo, the epicenter in Italy and still the deadliest place in Europe).

THe PREDICO score (multicenter study from Bologna-Italy) 10.1016/j.cmi.2020.08.003 should be take into consideration.

7) I am surprised the C-reactive protein did not find a place into the multivariate analysis, how can you explain instead the choice of N/L ratio?

Thanks

6. PLOS authors have the option to publish the peer review history of their article (what does this mean?). If published, this will include your full peer review and any attached files.

Reviewer #1: No

Reviewer #2: **Yes: **Lorenzo Roberto Suardi

---

## [Author Response · Author response to Decision Letter 0]

27 Jan 2021

Journal Requirements:

Thank you for this observation, the manuscript was edited according to the format required by PLOS ONE. 

2.Thank you for including your ethics statement: 'The protocol was approved by our scientific and ethics committees and conducted according to the principles of the Helsinki declaration.' 

(a) Please amend your current ethics statement to include the full name of the ethics committee/institutional review board(s) that approved your specific study. 

(b) Once you have amended this/these statement(s) in the Methods section of the manuscript, please add the same text to the “Ethics Statement” field of the submission form (via “Edit Submission”).

The protocol (protocol ID ABC-20-50) was approved by our local investigation and ethics (Comité de Ética en Investigación, Centro Médico ABC) and conducted according to the principles of the Helsinki declaration, this information was added to the manuscript and to the submission platform. 

3. In ethics statement in the manuscript and in the online submission form, please provide additional information about the patient records/samples used in your retrospective study. Specifically, please ensure that you have discussed whether all data/samples were fully anonymized before you accessed them and/or whether the IRB or ethics committee waived the requirement for informed consent. If patients provided informed written consent to have data/samples from their medical records used in research, please include this information.

The ethics committee waived the requirement for an informed consent. All the analyzed data was fully anonymized during the capturing process and was fully anonymized before access and analysis. This information was included in the manuscript.

4. Please note that PLOS does not permit references to “data not shown.” Authors should provide the relevant data within the manuscript, the Supporting Information files, or in a public repository. If the data are not a core part of the research study being presented, we ask that authors remove any references to these data.

Regarding the relevant data missing, the data has been added as a supplementary figure S5. 

5. Please amend your list of authors on the manuscript to ensure that each author is linked to an affiliation. Authors’ affiliations should reflect the institution where the work was done (if authors moved subsequently, you can also list the new affiliation stating “current affiliation:….” as necessary).

Affiliations have been revised and this has been corrected in the manuscript. 

6. One of the noted authors is a group or consortium [ARMII Study Group]. In addition to naming the author group, please list the individual authors and affiliations within this group in the acknowledgments section of your manuscript. Please also indicate clearly a lead author for this group along with a contact email address.

Mercedes Aguilar Soto is the lead author of the ARMII Study Group, all the authors are affiliated to Centro Médico ABC in México City. This information has been added to the manuscript. 

7. Please amend either the abstract on the online submission form (via Edit Submission) or the abstract in the manuscript so that they are identical.

Reviewers' comments:

Reviewer's Responses to Questions

Comments to the Author

1. Is the manuscript technically sound, and do the data support the conclusions?

Reviewer #1: Yes

Reviewer #2: Yes

2. Has the statistical analysis been performed appropriately and rigorously?

Reviewer #1: N/A

Reviewer #2: Yes

3. Have the authors made all data underlying the findings in their manuscript fully available?

Reviewer #1: No

Reviewer #2: Yes

4. Is the manuscript presented in an intelligible fashion and written in standard English?

Reviewer #1: Yes

Reviewer #2: Yes

5. Review Comments to the Author

Reviewer #1: Garcia-Gordillo conducted a retrospective study to investigate the new models to predict the need of IMV in patients with COVID-19. Although both NLR and IL-6 were well recognized as indicators of worse outcome in patients with covid-19, this study provided some useful information for clinical practices. I have some comments:

1) It’s problematic to use variables in the model at your own discretion which you describe in the method section. These variables should be determined base on standard statistic methods.

• Response: Using the development cohort, we performed univariate logistic regressions for IMV using all the variables mentioned above. We selected all variables that had a p value <0.1 and conducted a backwards stepwise multivariate logistic regression to find the variables that were independently associated with the requirement of IMV. After the selection of the optimal variables for the model, in order to ensure the model’s applicability in most settings, we checked for the variable’s availability in general settings. This was done by checking on 7 different general hospitals in Mexico City and its surroundings. The variables that were not available in more than 50% of the screened hospitals were deemed to be not readily available. We tested for similar variables using the Spearman correlation test in order to identify suitable surrogates. Thus, we developed two predictive models, one constructed with optimal variables and the other one with accessible surrogate variables. This information was included in the method section. 

2) The authors described in the abstract that “When compared with other similar 42 scores developed for the prediction of adverse outcomes in COVID-19, the COVID-IRS 43 scores proved to be superior in the prediction of IMV.” However, no comparisons were done and no p values were given in Table 2. Moreover, more scores such as APACHE-Ⅱand SOFA should be compared with the models.

• Response: Thank you for your observation, ROC curves comparisons have been added to Table 2. Even though comparisons with APACHE-II would be very valuable for our analysis, the EMR does not include all the variables required for the calculation of this score. We have calculated SOFA score and compared it with our score, nevertheless SOFA score includes a variable for IMV so scores are overestimated. In addition to this, due to the retrospective nature of our data, we did not distinguish patients who needed IMV on arrival from those who were intubated through their hospital stay. This was included in the manuscript.

3) The optimal cutoff value of these models should be shown in order to provide more information for clinical practices.

• Response: Optimal cutoff points in the validation cohort for the COVID-IRS-NLR score and the COVID-IRS-IL6 were >6 (S: 68.57%, E: 87.5%) and >5 (S: 72.86%, E: 81.67%), respectively. This information has been added to the manuscript. 

4) This study needs to be reviewed by a statistician.

• Response: The study was reviewed by a statistician as suggested by the reviewer

Reviewer #2: I am pleased to read a good work such this even if retrospective. I think some clarifications are needed in order to be published and in my opinion are minor revisions.

1) All the patients were evaluated to be, in case of need, "resuscitated"? I mean, It's important to clarify there was not a DNR order for some patients. 

• Response: Thank you for your comment, none of the patients included in our study had a DNR order at the time of admission or during hospitalization. This information was included in the manuscript. 

2) If possible, in order to identify the inflammatory phase, I would put the variable "time since symptoms onset to ED admission".

• Response: We appreciate the observation. In the development cohort the time from symptom onset to hospital admission was a median from 8 days (IQR 5.12) while in the validation cohort the median days from symptom onset to admission was of 7 days (IQR 6-11) (p= 0.63). This has been added to table 1.

3) All the patients came from community? Nosocomial or health care related infections?

• Response: SARS-Cov-2 infection was community acquired in all of our patients.

4) I would explain which treatment (corticosteroids, remdesivir, immunomodulators) have been given to the patients belonging to the 2 groups.

• Response: Thank you for this observation, several treatments were prescribed to the patients according to current guidelines, with no difference between groups. Lopinavir/ritonavir was given to 61.94% vs 60.11% of patients in the validation and development cohorts respectively (p=0.734); azithromycin was given to 76.77% vs 74.01% of patients in the validation and development cohorts respectively (p=0.560); hydroxychloroquine was given to 74.68% vs 81.92% of patients in the validation and development cohorts respectively (p=0.109); tocilizumab was given to 43.54% vs 42.68% of patients in the validation and development cohorts respectively (p=0.879); high dose glucocorticoids was given to 78.61% vs 76.67% of patients in the validation and development cohorts respectively (p=0.099). Remdesivir was not available in Mexico during our study period. This information was included in the manuscript and table 1. 

5) If not intubated, I would explain which kind of ventilatory support was given (nasal flow cannula, venturimask, CPAP, BiPAP)

• Response: Regarding the use of oxygen therapy in patients who did not require IMV, 65.2% in the development cohort and 69.2% in the validation cohort were treated with conventional nasal cannula (p=0.579). Three percent of the patients in the development cohort and none in the validation cohort required face tent (p=0.108). For non-rebreather mask the percentages were 8.97% for the development and 6.52 for the validation cohort (p=0.549). High-flow nasal cannula was required by 15% of the patients in the development cohort and 15.11% in the validation cohort (p=0.981). BIPAP/CPAP was used in 2.5% of the patients in the development cohort and 5.04% in the validation cohort (p=0.291). This information has been added to the manuscript. 

6) I woud cite in the discussion the Brescia-COVID Respiratory Severity Scale as the first score that was made worldwide out of China (Brescia has been, with Bergamo, the epicenter in Italy and still the deadliest place in Europe). The PREDICO score (multicenter study from Bologna-Italy) 10.1016/j.cmi.2020.08.003 should be take into consideration.

• Response: Thank you for this observation. Information regarding this scores has been added to the discussion and the PREDI-CO score was applied to our population and had a AUC of 0.704..

7) I am surprised the C-reactive protein did not find a place into the multivariate analysis, how can you explain instead the choice of N/L ratio?

• Response: The C-reactive protein was strongly correlated with the need for IMV in the univariate analysis, however, in the multivariate analysis, both the IL-6 and the NLR consistently outperformed the CRP’s predictive value, which was also a surprise for us.

---

## [Decision Letter · Decision Letter 1]

25 Feb 2021

COVID-IRS: a novel predictive score for risk of invasive mechanical ventilation in patients with COVID-19

PONE-D-20-32632R1

Dear Dr. Camiro-Zuñiga,

We’re pleased to inform you that your manuscript has been judged scientifically suitable for publication and will be formally accepted for publication once it meets all outstanding technical requirements.

Kind regards,

Antonio Palazón-Bru, PhD

Academic Editor

PLOS ONE

Additional Editor Comments (optional):

Reviewers' comments:

Reviewer's Responses to Questions

**Comments to the Author**

1. If the authors have adequately addressed your comments raised in a previous round of review and you feel that this manuscript is now acceptable for publication, you may indicate that here to bypass the “Comments to the Author” section, enter your conflict of interest statement in the “Confidential to Editor” section, and submit your "Accept" recommendation.

Reviewer #1: All comments have been addressed

Reviewer #2: All comments have been addressed

2. Is the manuscript technically sound, and do the data support the conclusions?

Reviewer #1: Yes

Reviewer #2: Yes

3. Has the statistical analysis been performed appropriately and rigorously? 

Reviewer #1: Yes

Reviewer #2: Yes

4. Have the authors made all data underlying the findings in their manuscript fully available?

Reviewer #1: Yes

Reviewer #2: Yes

5. Is the manuscript presented in an intelligible fashion and written in standard English?

Reviewer #1: Yes

Reviewer #2: Yes

6. Review Comments to the Author

Reviewer #1: My concerns have been addressed and I have no further questions. Thank authors for their frontline works and contributions to scientific field.

Reviewer #2: I think the Author had greatly answered to my comments and now the work could be of great interest to all the clinicians.

7. PLOS authors have the option to publish the peer review history of their article (what does this mean?). If published, this will include your full peer review and any attached files.

Reviewer #1: No

Reviewer #2: **Yes: **Lorenzo Roberto Suardi

---

## [Editor Report · Acceptance letter]

9 Mar 2021

PONE-D-20-32632R1 

COVID-IRS: a novel predictive score for risk of invasive mechanical ventilation in patients with COVID-19 

Dear Dr. Camiro-Zuñiga:

I'm pleased to inform you that your manuscript has been deemed suitable for publication in PLOS ONE. Congratulations! Your manuscript is now with our production department. 

Kind regards, 

on behalf of

Dr. Antonio Palazón-Bru 

Academic Editor

PLOS ONE